# Probabilistic Invariant Learning with Randomized Linear Classifiers

**Leonardo Cotta**
Vector Institute
leonardo.cotta@vectorinstitute.ai

**Gal Yehuda**
Technion, Haifa, Israel
ygal@cs.technion.ac.il

**Assaf Schuster**
Technion, Haifa, Israel
assaf@technion.ac.il

**Chris J. Maddison**
University of Toronto and Vector Institute
cmaddis@cs.toronto.edu

## Abstract

Designing models that are both expressive and preserve known invariances of tasks is an increasingly hard problem. Existing solutions tradeoff invariance for computational or memory resources. In this work, we show how to leverage randomness and design models that are both expressive and invariant but use less resources. Inspired by randomized algorithms, our key insight is that accepting probabilistic notions of universal approximation and invariance can reduce our resource requirements. More specifically, we propose a class of binary classification models called Randomized Linear Classifiers (RLCs). We give parameter and sample size conditions in which RLCs can, with high probability, approximate any (smooth) function while preserving invariance to compact group transformations. Leveraging this result, we design three RLCs that are provably probabilistic invariant for classification tasks over *sets, graphs, and spherical* data. We show how these models can achieve probabilistic invariance and universality using less resources than (deterministic) neural networks and their invariant counterparts. Finally, we empirically demonstrate the benefits of this new class of models on invariant tasks where deterministic invariant neural networks are known to struggle.

## 1 Introduction

A modern challenge in machine learning is designing model classes for invariant tasks that are both universal and resource-efficient. Consider designing an architecture for graph problems that is invariant under graph isomorphism. Ensuring that this class is universal is at least as hard as solving the graph isomorphism problem [5]. Similarly, permutation invariance universality comes at the cost of memory—the number of parameters scales with the sequence size [32].

Using randomness as a computational resource for algorithms is a fundamental idea in computer science. In their essence, randomized algorithms often provide simple solutions that use less memory or run in less time than their deterministic counterparts [30]. Since there is no free lunch, these gains come at the cost of accepting a probabilistic notion of correctness, *i.e.*, while deterministic algorithms always output correct answers, their randomized versions will be right only with high probability. Inspired by this, the key insight in our paper is using an external source of randomness to reduce time and space resources when learning an invariant task. To benefit from that, just as in randomized algorithms, we also need to introduce a probabilistic notion of invariance and expressive power. By guaranteeing invariance and universal approximation only with high probability, we show when it is possible to avoid the invariance-resource tradeoff.

37th Conference on Neural Information Processing Systems (NeurIPS 2023).

We introduce a class of models for binary classification tasks named *Randomized Linear Classifiers* (RLCs). In an RLC the external source of randomness is input to a neural network that generates a random linear classifier —independently from the input— which is then used to make a prediction. To guarantee correctness with high probability, we need to repeat this process for a sufficient number of times, *i.e.*, take multiple samples, and predict the majority of their output— this is called amplification. The key features of RLCs are both i) randomness independent from input and ii) linear transformation of the input. Combined, these features provide both theoretical and practical simplicity. On the practical side, RLCs can offload computation and preserve privacy in resource-constrained devices. We highlight the benefits of RLCs in such applications below.

a. *Online computation.* Suppose that our goal is to answer classification queries as quickly as possible, or that the queries arrive in an "online" fashion. By using RLCs, we can sample several linear classifiers "offline" (before observing new inputs), and then in the online phase simply take the majority of a few (different) linear classifiers for each input. This way, the majority of computations can be done offline rather than online —as usual in existing online learning settings.

b. *Private computation.* Suppose that we wish to peform inference with an RLC, but we only have a low-resource computer (*e.g.*, a smartwatch). A practical way of doing so is to perform the sampling on a remote server, then retrieve the answer. In our approach, the server can send the randomness, *i.e.*, the sampled linear classifier coefficients. There is no communication between the client and the server at all, and the client never sends the input —retaining its privacy. Moreover, since the client only needs to perform a few linear computations, it can be a low-resource computer.

The theoretical advantages of RLCs' features are explored in this work. We consider the required resources of an RLC to be universal and invariant for a given task as the number of parameters used by its neural network and the sufficient number of samples (amplification size). Then, we focus on establishing upper-bounds for these resources in different scenarios. More specifically, our contributions are three-fold:

i) We introduce the general class of Randomized Linear Classifiers (RLCs), presenting a universal approximation theorem coupled with a resource consumption characterization (Theorem 1). In specific, we show that any binary classification problem with a smooth boundary can be approximated with high probability by RLCs with at most the same number of parameters as a deterministic neural network. *This is a result of general purpose and holds for any invariant or non-invariant task.*

ii) We show how the problem of designing RLCs invariant to compact group transformations can be cast to designing invariant distributions of linear classifiers. As a result, we can leverage representation theorems in probability theory, such as de Finetti's [9], Aldous-Hoover 's[1] and Friedman's [13] to design invariant RLCs for tasks with set and graph data. Moreover, in Appendix B we also show a simple invariant model for spherical data using these ideas.

iii) Finally, we show both theoretically and empirically how invariant RLCs can succeed in tasks where deterministic architectures, such as Deep Sets and GNNs, cannot efficiently approximate. By introducing probabilistic notions of universality and invariance, our work establishes the first alternative to avoid the invariance-resource tradeoff with theoretical guarantees and empirical success.

## 1.1   Related Work

Here, we briefly review Coin-Flipping Neural Networks (CFNNs) and invariant representation learning —two core concepts that we build upon. Finally, we explain how Bayesian neural networks and RLCs belong to different learning paradigms.

**Coin-flipping neural networks.**   CFNNs were presented by [27], where the authors propose several deep learning architectures using randomness as part of their inference. A few (constructive) examples were shown in which randomness can help by reducing the number of parameters or the depth of the network. In this paper we focus on the case of learning randomized linear classifiers and its use in invariant learning. The invariant models were not explored in [27],

and to the best of our knowledge this is the first work studying randomized models for invariant learning. One specific CFNN model proposed in [27] is the Hypernetwork CFNN. In this model, the weights of a model are first randomly generated. Then, the computation of a deterministic neural network using these random weights takes place. In this work, we focus on the "extreme" case of this paradigm: we allow our model to use arbitrarily complex distributions to generate our final weights, but we insist that the deterministic part —using such weights— would be a *linear* function. That is, we push the computational complexity to the random part of the model. This gives rise to the class of models we study: Randomized Linear Classifiers (RLCs). The work of [27] considered a first version of RLCs, but did not provide any empirical results and their theoretical results hold only for single dimensional input (*i.e.*, in $\mathbb{R}$). Hence, to the best of our knowledge our work also establishes the first (general purpose) universal approximation theorem of (non-invariant) RLCs (*cf.* Section 2).

**Invariant representation learning.** Over the last years, there has been a growing interest in designing $G$-invariant architectures [4, 6, 18, 35]. In summary, we can often leverage an a priori knowledge that the function to be learned is invariant to the action of a group $G$ by forcing our model to also be invariant to it. That is, $G$-invariant architectures incorporate a known propriety of the target. Such a reduced model space is known to both empirically and theoretically achieve better generalization [4, 6, 11, 19]. Existing works study the $G$-invariant function space and how to parameterize it with neural networks. Despite some empirical success, invariance guarantees often come at unreasonable prices. For instance, CNNs are guaranteed to be invariant to translations only with infinite grids (images) [19]. Regarding permutation invariance, Deep Sets is always invariant, but to remain universal it requires a hidden layer as large as the input set [32]. Finally, GNNs, as any other computationally efficient graph isomorphism-invariant architecture, cannot achieve universal approximation. More specifically, achieving graph isomorphism invariance and universal approximation is at least as hard as solving the graph isomorphism problem [5]. In contrast to these results, our work shows that such tradeoffs are usually restricted to the design of deterministic architectures. By accepting probabilistic notions of invariance and universality, we show that under mild data generation conditions we can circumvent these invariance-resource tradeoffs.

**Bayesian neural networks.** Bayesian Neural Networks (BNNs) incorporate Bayesian inference into both training and inference of neural networks. In short, BNNs explicitly assign a prior to the parameters and perform inference with a posterior. BNNs differ from RLCs in how randomness is used. BNNs explicitly model prior and posterior distributions over the weights. This is usually leveraged in the context of confidence estimation. On the other hand, RLCs do not model a prior or a posterior distribution over the parameters. Rather, RLCs take samples from a model that uses external randomness. In their essence, BNNs and RLCs are trying to achieve different goals. While BNNs are modeling uncertainty over decisions, RLCs are trying to output the correct target with the highest probability they can.

## 2 Randomized Linear Classifiers

We start by introducing the general class of Randomized Linear Classifiers. Due to our universal approximation result, this can be seen as the analogous of a multi-layer perceptron in the context of randomized linear models. In the next sections we will introduce the analogous of $G$-invariant models.

For notation simplicity, we denote the random variable of a value with bold letters, *e.g.*, a random variable $\mathbf{x}$ would have a realization $x$. We consider binary classification tasks where the input data $\mathbf{x}$ is supported on $\mathcal{X} := \mathrm{supp}(\mathbf{x}) \subseteq \mathbb{R}^d$ and labeled according to a true labeling function $y \colon \mathcal{X} \to \{-1, 1\}$. Therefore, a binary classification task can be summarized by the tuple $(\mathbf{x}, y)$.

Given a source of randomness $\mathbf{u}$ supported on $\mathcal{U} := \mathrm{supp}(\mathbf{u}) \subseteq \mathbb{R}^{d_u}$, and a neural network $f_\theta \colon \mathbb{R}^{d_u} \to \mathbb{R}^{d+1}$ parameterized by $\theta \in \mathbb{R}^d$, an RLC predicts $y(x)$ for $x \in \mathcal{X}$ according to

$$\mathrm{sgn}(\langle \mathbf{a}_\theta, x \rangle - \mathbf{b}_\theta) := \mathrm{sgn}(\langle f_\theta(\mathbf{u})_{1:d}, x \rangle - f_\theta(\mathbf{u})_{d+1}). \tag{1}$$

From above, one can see the coefficients $(\mathbf{a}_\theta, \mathbf{b}_\theta)$ of the linear classifier given by the pushforward measure $f_\theta \# \mathbf{u}$. Finally, unless otherwise stated, we assume $f_\theta$ to be a multi-layer perceptron with ReLU activations.

Analogously to randomized algorithms for decision problems, we can take multiple samples using Equation (1) and make a final prediction using their majority. To simplify notation, let us define the function of our final prediction taking $m$ samples as

$$\mathbf{y}_\theta^{(m)}(x) := \mathrm{maj}(\{\mathrm{sgn}(\langle \mathbf{a}_\theta^{(j)}, x \rangle - \mathbf{b}_\theta^{(j)})\}_{j=1}^m) = \mathrm{maj}(\{\mathrm{sgn}(\langle f_\theta(\mathbf{u}^{(j)})_{1:d}, x \rangle - f_\theta(\mathbf{u}^{(j)})_{d+1}\}_{j=1}^m),$$
(2)

where each $\mathbf{u}^{(j)} \sim_{\mathrm{i.i.d}} P(\mathbf{u})$. Note that as we take more samples, we approach the mode of the distribution of linear predictions for a given $x$. We define this value as the limiting classification of $x$, which implies the existence of the limiting classifier $\mathcal{F}_\theta \colon \mathbb{R}^d \to \{-1, 1\}$ with

$$\mathcal{F}_\theta(x) := \lim_{m \to \infty} \mathrm{maj}(\{\mathrm{sgn}(\langle \mathbf{a}_\theta^{(j)}, x \rangle - \mathbf{b}_\theta^{(j)})\}_{j=1}^m) = \underset{\hat{y} \in \{-1,1\}}{\mathrm{argmax}} \, P(\mathrm{sgn}(\langle \mathbf{a}_\theta, x \rangle - \mathbf{b}_\theta) = \hat{y}).$$
(3)

At this point, it is worth noting that although the random prediction (Equation (1)) is a linear function, its limiting classifier (Equation (3)) can induce an arbitrarily non-linear boundary. In fact, we leverage this notion to define the probabilistic version of universality. **We say that an RLC is universal in a probabilistic sense if its limiting classifier (Equation (3)) is universal.** It is easy to see that models making deterministic predictions collapse the notions of probabilistic and exact universality. On the other hand, for RLCs to be probabilistic universal it suffices to produce predictions that are biased towards the desired answer on every input, *i.e.*, $P(\mathbf{y}_\theta^{(m)}(x) = y(x)) > 0.5, \forall x \in \mathcal{X}$.

In practice, we will make random predictions and therefore we need to capture a sufficient sample size $m$ such that we are close to the limiting classifier with high probability. An important measure to understand how large $m$ needs to be is what we call the minimum bias of the RLC. We denote it by $\varepsilon$ and define it as the infimum of the total variation distances between the random predictions of every input $x$ and a random variable $\mathbf{r}$ with Radamacher distribution. That is,

$$\varepsilon := \inf\{d_{\mathrm{VD}}(\mathbf{y}_\theta^{(m)}(x), \mathbf{r}) \colon x \in \mathcal{X}\}.$$

Put into words, $\varepsilon$ captures a lower-bound on how close to a random prediction the RLC can be. And, naturally, the closer it is to a random prediction the larger $m$ needs to be such that the majority converges to the limiting prediction. Therefore, our universal approximation result needs to capture how many parameters an RLC needs to converge to correct answers and how fast, *i.e.*, with respect to $m$, it does. Before we proceed with our results on $m$, $p$ and the universality property, we need to define the general class of binary classification tasks we consider.

**Assumption 1** (Tasks with smooth separators). *We say that a binary classification task $(\boldsymbol{x}, y)$ admits a smooth separator if there exists a neural network $s \colon \mathbb{R}^d \to \mathbb{R}$ such that $y(x) = \mathrm{sgn}(s(x)), \forall x \in \mathrm{supp}(X)$.*

Note that due to the universal approximation ability of neural networks, the set of tasks satisfying Assumption 1 is quite large. Moreover, **Assumption 1 is a sufficient but not necessary condition**. We will later prove universality for a wide class of graph problems and we believe many others exist. For now, let us finally state our result on the resource consumption and universal approximation of RLCs in tasks with smooth separators.

**Assumption 2** (Absolutely continuous randomness source). *We say that the randomness source $\boldsymbol{u}$ is absolutely continuous if at least one of its coordinates $\boldsymbol{u}_i, 1 \leq i \leq d_{\boldsymbol{u}}$ has an absolutely continuous marginal distribution.*

**Theorem 1** (Resource consumption of universal RLCs). *Let $(\boldsymbol{x}, y)$ be a binary classification task that admits a smooth separator as in Assumption 1. Then, there exists an RLC with neural network $f_{\theta^\star}$ and absolutely continuous randomness source $\boldsymbol{u}$ (Assumption 2) that is universal in the limit, i.e.,*

$$\mathcal{F}_{\theta^\star}(x) = y(x), \forall x \in \mathcal{X},$$

*and makes random predictions that are correct with probability*

$$P(\mathrm{maj}(\{\mathrm{sgn}(\langle \boldsymbol{a}_{\theta^\star}^{(j)}, x \rangle - \boldsymbol{b}_{\theta^\star}^{(j)})\}_{j=1}^m) = y(x)) > 1 - \exp\{-2\epsilon^2 m^2\},$$

*where $\epsilon$ is the minimum bias of $\mathcal{F}_{\theta^\star}$.*

*Further, if $p^\dagger$ is the number of parameters used by a deterministic neural network with one hidden layer to achieve zero-error in the task, $f_\theta$ has at most*

$$p \leq p^\dagger + \mathcal{O}(1) \text{ parameters.}$$

The complete proof is in Appendix A Note that Theorem 1 does not assume or explore any invariance in the task. It is a general universality result that we will leverage in our study of invariant RLCs, but of independent interest. One way to interpret Theorem 1 is as a probabilistic version of the classical universal approximation theorem of multi-layer perceptrons [8]. The result on the number of parameters is important to make it clear that RLCs do not need to find a solution more complex than one given by a deterministic model in the supervised learning task.

Until now, we have established the expressive power of RLCs, *i.e.*, that, under mild assumptions, they can be as expressive and resource-efficient as deterministic neural networks. But, there is still the general question: When are RLCs more resource-efficient than deterministic neural networks? In [27] it was constructed an specific task where an RLC (although the authors do not name it as such) uses a constant number of parameters, while a deterministic neural network would need a number of parameters that grows with the input dimension. Here, we are interested in more general settings. In the next section we will show a large class of invariant tasks where invariant RLCs are more resource-efficient than their invariant deterministic neural network counterparts.

## 3  $G$-invariant Randomized Linear Classifiers

Now, we turn our focus to binary classification tasks that are invariant to compact group transformations. That is, let $G$ be a compact group with an action of $g \in G$ on $x \in \mathcal{X}$ denoted by $g \cdot x$. A task $(\mathbf{x}, y)$ is $G$-invariant if $y(x) = y(g \cdot x), \forall x \in \mathcal{X}, \forall g \in G$. It is easy to see that an RLC that perfectly classifies the task in the probabilistic notion, *i.e.*, $\mathcal{F}_{\theta^\star}(x) = y(x), \forall x \in \mathcal{X}$, will also be probabilistic $G$-invariant, *i.e.*, $\mathcal{F}_{\theta^\star}(x) = \mathcal{F}_{\theta^\star}(g \cdot x), \forall x \in \mathcal{X}, \forall g \in G$. Thus, if we know that a task is $G$-invariant, can we restrict our model class to $G$-invariant RLCs? At first, the answer is not obvious. We need a design principle that guarantees a search space with only $G$-invariant solutions while at least one of them has zero error. In Theorem 2 we present the result that guides our design of invariant RLCs.

**Theorem 2** ($G$-invariant RLCs)**.** *Let $(\boldsymbol{x}, y)$ be a $G$-invariant task with a smooth separator as in Assumption 1. Then, the set of RLCs with a $G$-invariant distribution in the classifier weights, i.e.,*

$$\boldsymbol{a}_\theta \overset{d}{=} g \cdot \boldsymbol{a}_\theta, \forall g \in G,$$

*and absolutely continuous randomness source (cf. Assumption 2) is both probabilistic $G$-invariant and universal in $(\boldsymbol{x}, y)$. That is,*

$$\mathcal{F}_\theta(x) = \mathcal{F}_\theta(g \cdot x), \forall x \in \mathcal{X}, \forall g \in G, \forall \theta \in \mathbb{R}^p,$$

*and*

$$\exists \theta^\star \in \mathbb{R}^p \colon \mathcal{F}_{\theta^\star}(x) = y(x), \forall x \in \mathcal{X}, \forall g \in G.$$

The complete proof is in Appendix A From above, we now know that designing universal RLCs for invariant tasks can be cast to the problem of designing universal $G$-invariant distributions —together with a separate (possibly dependent) bias distribution. This design principle differs fundamentally from deterministic invariant representation learning. Here, instead of designing architectures that are $G$-invariant, we wish to design architectures that induce a $G$-invariant distribution. That is, the RLC neural network $f_\theta$ is not itself $G$-invariant.

Next, we will define a very useful assumption about the data generation process. It will allow us to leverage representation results in probability theory to both design universal and efficient $G$-invariant distributions and allow for variable-size set and graph data.

**Assumption 3** (Infinitely $G$-invariant data)**.** *Let $G_\infty$ be a homomorphism of $G$ into the composition of homomorphisms of $G$ in all finite dimensions, e.g., if $G$ is the set of permutations of $\{1, \ldots, d\}$, $G_\infty$ is the set of permutations of $\mathbb{N}$. We say that a task $(\boldsymbol{x}, y)$ has infinitely $G$-invariant data if there exists an infinite sequence of random variables $(\boldsymbol{x}_i)_{i=1}^\infty \overset{d}{=} g_\infty \cdot (\boldsymbol{x}_i)_{i=1}^\infty, \forall g_\infty \in G_\infty$, where $\boldsymbol{x} \overset{d}{=} (\boldsymbol{x}_i)_{i \in S}$ for $S \in \binom{\mathbb{N}}{d}$, with $\binom{\mathbb{N}}{d}$ being the set of all $d$-size subsets of $\mathbb{N}$.*

Although the above might seem unintuitive, the representation theorems also help us make sense of them. The data generation process will take the same form as our linear weights distribution. As we will see, for sets it means that items are sampled i.i.d. given a common latent factor of the set [9]. In graphs edges are also generated i.i.d. given latent factors of their endpoints and the graph [1].

**Proposition 1** (Infinitely $G$-invariant RLCs). *Let $(\boldsymbol{x}, y)$ be a $G$-invariant task with infinitely $G$-invariant data (Assumption 3) with a smooth separator as in Assumption 1. Then, the set of RLCs with an infinitely $G$-invariant distribution in the linear classifier weights, i.e., as in Assumption 3 $(\boldsymbol{a}_{\theta i})_{i=1}^{\infty} \overset{d}{=} g_{\infty} \cdot (\boldsymbol{a}_{\theta i})_{i=1}^{\infty}, \forall g_{\infty} \in G_{\infty}$, where $\boldsymbol{a}_{\theta} \overset{d}{=} (\boldsymbol{a}_{\theta i})_{i \in S}$, for $S \in \binom{\mathbb{N}}{d}$, and absolutely continuous randomness source (cf. Assumption 2) is probabilistic $G$-invariant and universal for $(\boldsymbol{x}, y)$ as in Theorem 2.*

The complete proof is in Appendix A Next, we design $G$-invariant RLCs drawing from results in the representation theory of probability distributions. We consider tasks with set and graph data. In Appendix B we also consider spherical data. For each, we derive conditions for $G$-invariance and universality, highlighting their resource gain when compared to their deterministic counterpart.

## 3.1 RLCs for set data

Here we consider tasks that are invariant to permutations, *i.e.*, our input is a set[1] of vectors. More formally, we let our input be supported on $\mathcal{X} := (\mathbb{R}^k)^d$[2] and $G := \mathrm{Sym}([d])$ be the group of permutations of $[d] := \{1, \ldots, d\}$. Then, for an input $x \in \mathcal{X}$ the action $g \cdot x$ is given by permuting the $d$ vectors of size $k$ according to $g$.

The key insight to design RLCs for set data is leveraging the classic de Finetti's theorem [9]. In summary, de Finetti showed how any infinite sequence of exchangeable random variables can be expressed as an infinite sequence of i.i.d. random variables conditioned on a common latent measure. Now, when Assumption 3 is true, de Finetti tells us that we can sample our weights $\mathbf{a}_{\theta}$ by first sampling a common noise and use it as input to the same function, which generates the linear weights together with an independent noise in an i.i.d. manner. Note that the bias does not have the exchangeability property. Thus, it needs to receive the same shared noise as the set but use a different function, since it is not necessarily sampled in the same way. Next, we formalize these notions by defining the class of Randomized Set Classifiers (RSetCs) and characterizing its universal approximation power together with resource consumption.

**Definition 1** (Randomized Set Classifiers (RSetCs)). *A Randomized Set Classifier (RSetC) uses two neural networks $f_{\theta_f} : \mathbb{R}^2 \to \mathbb{R}^k$ and $g_{\theta_g} : \mathbb{R}^2 \to \mathbb{R}$ together with $d + 2$ sources of randomness: $\boldsymbol{u}$, $(\boldsymbol{u}_i)_{i=1}^d$, and $\boldsymbol{u}_b$. The random linear classifier coefficients are generated with*

$$\boldsymbol{a}_{\theta i}^{(k)} \overset{d}{=} f_{\theta_f}(\boldsymbol{u}, \boldsymbol{u}_i) \text{ and } \boldsymbol{b}_{\theta} \overset{d}{=} g_{\theta_g}(\boldsymbol{u}, \boldsymbol{u}_b),$$

*where $\boldsymbol{a}_{\theta i}^{(k)}$ is the $i^{th}$ chunk of $k$ random linear weights, i.e., the weights multiplying the $i^{th}$ vector.*

**Proposition 2** (Universality and resource consumption of RSetCs). *Let $(\boldsymbol{x}, y)$ be a permutation-invariant task with a smooth separator as in Assumption 1 and infinitely permutation-invariant data as in Assumption 3. Then, RSetCs as in Definition 1 with absolutely continuous randomness source (cf. Assumption 2) are probabilistic $G$-invariant and universal for $(\boldsymbol{x}, y)$ (as in Theorem 2). Further, the number of parameters needed by RSetCs in this task will depend only on the smallest finite absolute moments of the weight and bias distributions in the zero-error solution, i.e., the ones given by $f_{\theta_f^{\star}}(\boldsymbol{u}, \boldsymbol{u}_i)$ and $g_{\theta_g^{\star}}(\boldsymbol{u}, \boldsymbol{u}_b)$.*

The complete proof is in Appendix A We highlight that Proposition 2 is combining de Finetti's theorem [9], Kallenberg's noise transfer theorem [16] and recent results on the capacity of neural networks to generate distributions from noise [34].

**What is the practical impact of Assumption 3 in set tasks?** Assumption 3 implies that an input set $\mathbf{x}$ is generated by first sampling some latent variable $\Theta \sim \nu$ which is then used to sample items in an i.i.d. manner $\mathbf{x}_1, \mathbf{x}_2, \ldots |\sim_{\Theta \text{i.i.d.}} P_{\Theta}$. We can make sense of this with an example in e-commerce systems. If the set represents items in a shopping cart, the latent factor $\Theta$ can be thought of as a summary of the user's shopping behavior. Overall, this assumption is not appropriate if there could be mutually exclusive items in the set.

**When are RSetCs better than Deep Sets?** We contrast the results of RSetCs with Deep Sets [35] due to its universality and computational efficiency [32]. Overall, other deterministic set models

---

[1]If the support is discrete we shall consider multisets instead.

[2]Note that this does not change the reference to $d$ in previous results.

are either variations of Deep Sets that inherent the same properties [20, 24] or are computationally inefficient [23]. Apart from universality, it is important to note from Proposition 2 how the number of parameters used by the RSetC does not depend on the set size. Instead, its resource consumption is related to how smooth the distribution of linear classifiers weights and bias are. Since all the weights have the same distribution (conditioned on **u**), the set size is not relevant. This comes in contrast to Deep Sets, which as shown in [32] only achieves universality when the hidden-layer and the input set have the same size.

Moreover, Deep Sets is known to have poor set size generalization abilities, *i.e.*, when the test set contains larger sets than the training set [31]. In RSetCs, although the number of parameters does not depend on the set size, the distribution of linear coefficients can use this information. Overall, the gain in robustness to set size might come from RSetCs performing simple linear transformations of the input. That is, for small changes in the set size, the output is not expected to change abruptly. This opposes to Deep Sets, where the representations of the items are aggregated and input to a neural network. Thus, any small addition to the set can arbitrarily change the output. Although we cannot always guarantee size generalization for RSetCs, we give empirical evidence in Section 4 that they do seem to be a lot more robust than Deep Sets.

## 3.2  RLCs for graph data

Here we consider tasks that take graphs as input. Given a (possibly weighted) graph $G = (V, E)$ with $d$ vertices and $M$ edges, we take as input a vector $x$ of size $d^2$ representing the vectorized adjacency matrix of $G$. Our invariance is then to graph isomorphism. That is, we again have the permutation group $G := \text{Sym}([d])$, but now with a different action. Here $g \cdot x$ jointly permutes the rows and columns of the adjacency matrix corresponding to $x$. This way, $x$ and $g \cdot x$ represent isomorphic graphs. We will then assume that our graph tasks are $G$-invariant, *i.e.*, two isomorphic graphs $x, g \cdot x$ will always have the same label.

Now, we can leverage Aldous-Hoover's theorem, the analogous of de Finetti's theorem for joint exchangeability in 2-dimensional arrays. It gives us a model, invariance and universality results analogous to the ones for RSetCs.

**Definition 2** (Randomized Graph Classifiers). *A Randomized Graph Classifier (RGraphC) uses two neural networks $f_{\theta_f} \colon \mathbb{R}^4 \to \mathbb{R}$ and $g_{\theta_g} \colon \mathbb{R} \to \mathbb{R}$ together with $d^2 + d + 1$ sources of randomness: $\boldsymbol{u}$, $(\boldsymbol{u}_i^{(n)})_{i=1}^d$, $(\boldsymbol{u}_{ij}^{(e)})_{i,j \in [d]}$, and $\boldsymbol{u}_b$. The random linear classifier coefficients are generated with*

$$\boldsymbol{a}_{\theta ij} \overset{d}{=} f_{\theta_f}(\boldsymbol{u}, \boldsymbol{u}_i^{(n)}, \boldsymbol{u}_j^{(n)}, \boldsymbol{u}_{ij}^{(e)}), \text{ and } \boldsymbol{b}_\theta \overset{d}{=} f_{\theta_f}(\boldsymbol{u}, \boldsymbol{u}_b),$$

*where $\boldsymbol{a}_{\theta ij}$ is the random linear weight multiplying the entry of $x$ that corresponds to the entry in row $i$ and column $j$ of the graph's adjacency matrix.*

Note that we can use the same proof of Proposition 2 and have an equivalent result for RGraphCs. However, since graphs are discrete objects the smooth separator assumption (*cf.* Assumption 1) might not be so easily satisfied. To address this, we define a sufficiently large class of graph models that RGraphCs can approximate.

**Definition 3** (Inner-product decision graph problems). *Let $G = (V, E)$ be a graph over $d$ vertices with vectorized adjacency matrix $x \in \mathcal{X} := \text{supp}(\boldsymbol{x}) \subseteq \mathbb{R}^{d^2}$. We say that a graph property $y \colon \mathcal{X} \to \{-1, 1\}$ is an* inner product verifiable property *if there exists a set of vectors $S \subseteq \mathbb{R}^{d^2}$ and a constant $b \in \mathbb{R}$ such that:*

- *For all graphs $x^+ \in \mathcal{X}$ satisfying the property $y(x^+) = 1$, there exists $s \in S$, such that $\langle s, x^+ \rangle \geq b$;*

- *For all graphs $x^- \in \mathcal{X}$ not satisfying the property $y(x^-) = -1$, we have $\langle x^-, s' \rangle < b, \forall s' \in S$.*

As an example of such property, consider connectivity. The set $S$ above is defined to be the set of all binary vectors representing adjacencies of spanning trees. The threshold in this case would be $d - 1$. If $G$ is connected, then there exists a spanning tree for $G$. When taking the inner product of $x$ and the vector $s \in S$ representing the spanning tree, the result would be the number of edges in the tree, which is $d - 1$. In contrast, if $G$ is not connected, it does not have a spanning tree. This means that for all vectors $s'$ in $S$, the inner product $\langle s', x \rangle$ is less than $d - 1$. We can use the same logic to show that properties arising from NP-complete problems such as independent set, or simple ones that GNNs

cannot approximate, such as diameter, girth and connectivity are all encompassed by Definition 3. We are now ready to state the probabilistic invariance and universality result for RGraphCs.

**Theorem 3.** *Let $(\boldsymbol{x}, y)$ be a graph isomorphism-invariant task that either i) has a smooth separator as in Assumption 1 or ii) is an inner-product decision graph problem as in Definition 3. Further, the task has infinitely graph isomorphism-invariant data as in Assumption 3. Then, there exists an RGraphC as in Definition 2 with absolutely continuous randomness source (cf. Assumption 2) that is probabilistic G-invariant and universal for $(\boldsymbol{x}, y)$ (as in Theorem 2). Further, the number of parameters of this RGraphC will depend only on the smallest finite absolute moments of its weight and bias distributions, i.e., the ones given by $f_{\theta_f}(\boldsymbol{u}, \boldsymbol{u}_i)$ and $g_{\theta_g}(\boldsymbol{u}, \boldsymbol{u}_b)$.*

The complete proof is in Appendix A The main insight behind Theorem 3 is using techniques from randomized algorithms to derive the universality of RLCs in the graph tasks from Definition 3. The other results follow the same line as Proposition 2 while replacing de Finetti's with Aldous-Hoover's theorem.

**What is the practical impact of Assumption 3 in graph tasks?** Janson and Diaconis [15] showed that Assumption 3 is equivalent to having the input graphs sampled from a graphon model. For the reader unfamiliar with graphons, this class encompasses from simple random graph models like $G(n, p)$ [12], to modern matrix factorization methods in recommender systems [29]. Moreover, this assumption has been recently used to design representations that are invariant to graph size, see [3].

**When are RGraphCs better than GNNs?** We contrast the results of RGraphCs with GNNs[17, 22, 26] due its widespread use, empirical success and computational efficiency. A known issue with GNNs is its inability to approximate simple tasks such as graph connectivity [14]. This comes from the fact that GNNs cannot distinguish simple non-isomorphic graphs such as any pair of $d$-regular graphs. As such, GNNs cannot universally approximate tasks that assign different labels to any of such pairs. In contrast, RGraphCs can approximate any problem that either has a smooth boundary or can be tested with an inner product as in Definition 3.

Note that we can always define a task following Definition 3 that distinguishes between a specific graph and any other non-isomorphic input. Thus, unlike deterministic models, the set of tasks RGraphCs can solve is not attached to solving the graph isomorphism problem. It is simply attached to whether the decision problem can be tested with an inner product. Note that this does not imply that our model solves the graph isomorphism problem. To do so, we would need to test on an exponential number of tasks. The main implication of this observation is that **the expressive power of probabilistic graph models is not attached to the ability to distinguish non-isomorphic graphs, but to the task's complexity.** Thus, we are not doomed to fail at simple tasks like GNNs. Finally, note that GNNs aggregate messages using Deep Sets (or some variation of it). Thus, its approximation power, even for graphs that it can distinguish, requires a number of parameters that grows with the graphs' number of vertices.

Finally, we highlight that there exist more expressive, but still non-universal, models such as higher-order GNNs [22] and subgraph GNNs [2, 7]. However, they suffer from a heavy use of computational resources and are restricted to smaller, but existent, set of non-isomorphic graphs it cannot distinguish. As such, they suffer from the same aforementioned problems as GNNs.

## 4 Experiments

Until now, our theory characterized the model space of invariant RLCs. It is natural to wonder whether such guarantees translate into practice. Concretely, we focus on investigating three questions related to our theoretical contributions.

**Q1.** Are RSetCs more parameter-efficient than Deep Sets in practice? (*cf.* Proposition 2)
**Q2.** Are RSetCs more robust than Deep Sets with out-of-distribution data? (*cf.* discussion in Section 3.1)
**Q3.** Can RGraphCs efficiently solve tasks that GNNs struggle in practice? (*cf.* Theorem 3)

**Sorting task.** To address **Q1** and **Q2**, we consider the sorting task proposed in [32]. Our input is given by $\mathbf{x}, \operatorname{supp}(\mathbf{x}) = \mathbb{R}^d, \mathbf{x}_i \sim \mathcal{N}(0, 1)$ for some odd number of dimensions $d$ and the labeling function by $y(\mathbf{x}) = 1$ if for a vector $w, w_i := (-1)^{i+1}$ we have $w^T \texttt{sort}(\mathbf{x}) \geq 0$ or $-1$ otherwise. We chose this task since it was shown in [32] that Deep Sets needs a hidden-layer at least as large as the input set ($d$) to approximate it with zero error. Thus, to showcase the parameter efficiency of

RSetCs in practice (*cf.* Proposition 2) we test both models in this task, while fixing the hidden layer at 5 on both. To make the results comparable, we train RSetCs with a single hidden layer (in both networks) and Deep Sets with a single hidden layer in each of its networks as well. Finally, to answer **Q2** we test both models on sets twice as large as the ones used in training.

**Sign task.** We would also like to investigate **Q1** in a setting where Deep Sets is not supposed to struggle. That is, we would like to know whether RSetCs can be more parameter-efficient in tasks that are (supposedly) easy for Deep Sets. For this, we consider the sign task $y(\mathbf{x}) := \mathrm{sgn}(\sum_{i=1}^{d} \mathrm{sgn}(\mathbf{x}_i))$ with $\mathbf{x}_i \sim N(0, 1)$. Note that the networks in Deep Sets would need to simply learn the sign function and thus its parameters should not depend on the set size. To showcase the parameter efficiency of RSetCs, we consider Deep Sets with one and two hidden layers (in both networks) while keeping RSetCs with a single hidden layer. The rest of the experimental setup follows the sorting task verbatim.

**Connectivity task.** We chose the task of deciding whether a graph sampled from a $G(n, p)$ model with $p = 1.1 \cdot \log(n)/n$ is connected or not. Unlike RGraphCs, GNNs provably cannot approximate this simple task [14]. By comparing the performance of GNNs and RGraphCs in this task, we can empirically verify Theorem 3. Moreover, in contrast to GNNs, Theorem 3 tells us that the required number of parameters in RGraphCs is not attached to the graph size. Hence, we also test the performance of both models in this task while increasing the size of the input graphs and keeping hidden layer sizes fixed to 2 in both models. To make results comparable, we use a GNN with 3 layers and an RGraphC with 3 hidden layers of size in each network. Since in each GNN layer there are two network layers, the parameter sizes in the GNN and in the RGraphC are comparable.

**Experimental setup.** We used the Deep Sets architecture as proposed in the original paper [35]. For the GNN, we used the GIN [33] architecture, which completely captures the GNN properties mentioned in Section 3.2. Finally, to give perspective on how bad a model is performing, we also contrast the results with a constant classifier, *i.e.*, how a classifier that always predicts the most common class performs. All models were trained with Pytorch [25] using Adagrad [10] for a maximum of 1000 epochs with early stopping. The reported results in Figures 1 and 2b are with respect to five runs. The training sets consisted of 1000 examples, while the the validation and test sets contained 100 examples. We detail the hyperparameters and their search in Appendix C.

**A1 (RSetC parameter efficiency).** In Figure 1a we see the (test set) results for RSetCs and Deep Sets when varying the size of the input for a fixed hidden layer. We note that even in the first task, $d = 5$, when Deep Sets is supposed to approximate the task well it fails to generalize. Overall, for all input sizes we can see RSetCs consistently outperforming Deep Sets. On the other hand, in most of the tasks Deep Sets performs similarly or even worse than a constant classifier. Finally, in the sign task from Figure 2a we see that the performance of a single hidden-layer RSetC is closer to a two hidden-layer Depe Sets, showing its parameter efficiency even in tasks where Deep Sets is supposed to efficiently succeed. Thus, we can verify Theorem 2 and Proposition 2 in practice.

**A2 (RSetC out-of-distribution robustness).** We can see in Figure 1b how RSetCs provide consistently better results than Deep Sets when tested in sets twice the size as the ones used in training. This provides evidence to the size generalization discussion in Section 3.1. We can see that RSetCs perform even similarly to the in-distribution case, while Deep Sets is consistently worse than a constant classifier. Moreover, the high variance in Deep Sets' results confirm our observation in Section 3.1 about its sensitivity.

**A3 (RGraphC universality and parameter efficiency).** In Figure 2b we can see the results for RGraphCs vs. GNNs in the connectivity task. We note how the lack of expressive power coupled with the parameter inefficiency of GNNs is reflected in its poor performance, that decreases as the input size increases. In contrast, Theorem 3 is confirmed by RGraphCs' better performance even when the input size is increased (and the number of parameters is fixed). Finally, just as in Deep Sets we note a very high variance in the GNN results. This is often observed in tasks where there is no continuous features in the input, see *e.g.*, [7].

## 5    Conclusions

Our work established the first class of models that leverages randomness to achieve universality and invariance in binary classification tasks. We combined tools from randomized algorithms and probability theory in our results. By leveraging this new principle, we were able to present resource-

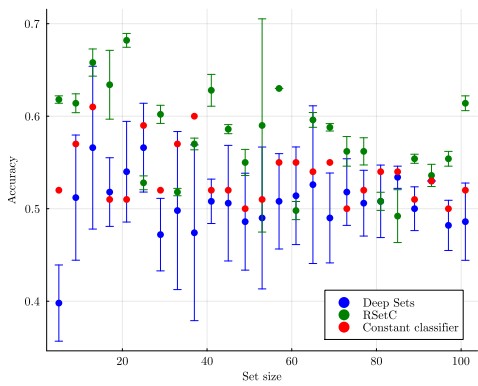
(a) Results with in-distribution test set.

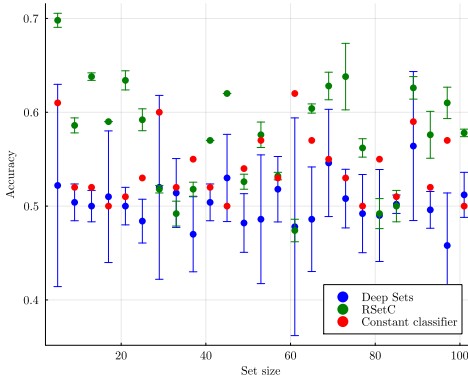
(b) Results using a test set containing sets twice the size of the ones used in training.

Figure 1: Results for the sorting task. We use a fixed hidden layer size of 5 in both models, while varying the training input size. The mean accuracy, together with standard deviation values, is reported using five runs.

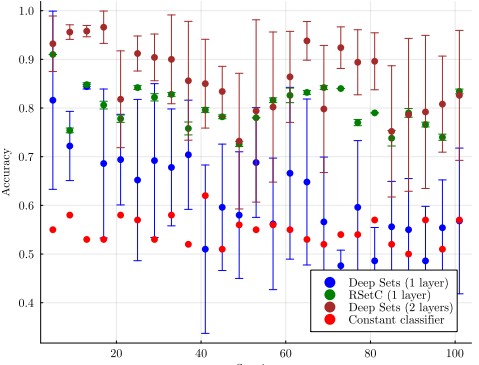
(a) Results for the sign task. We use a fixed hidden layer size of 5 in both models, while varying the training input size. The mean accuracy, together with standard deviation values, is reported using five runs.

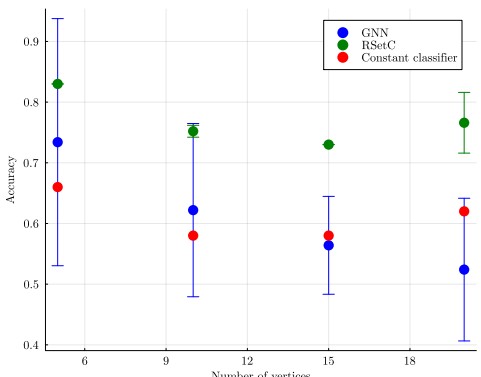
(b) Results for the connectivity task. We use a fixed hidden layer size of 2 in both models, while varying graph size. We report mean accuracy and standard deviation over five runs.

Figure 2: Results for the sign (set) and the connectivity (graph) tasks.

efficient models that, under mild assumptions, are universal and invariant in a probabilistic sense. This work can be extended in many different ways, *e.g.*, designing architectures for other group invariances, such as $\mathrm{SO}(n)$ or designing specific optimization procedures for RLCs. It is also interesting to understand the practical benefit of RLCs and their invariant counterparts in real-world tasks.

## Acknowledgements

We acknowledge the support of the Natural Sciences and Engineering Research Council of Canada (NSERC), RGPIN-2021-03445 and of the Hasso Plattner Institute. L. Cotta is funded in part, by a postdoctoral fellowship provided by the Province of Ontario, the Government of Canada through CIFAR, and companies sponsoring the Vector Institute. This work was done in part, when G. Yehuda was visiting the Simons Institute for the Theory of Computing.

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

# A  Proofs

**Theorem 1** (Resource consumption of universal RLCs). *Let $(x, y)$ be a binary classification task that admits a smooth separator as in Assumption 1. Then, there exists an RLC with neural network $f_{\theta^\star}$ and absolutely continuous randomness source $\boldsymbol{u}$ (Assumption 2) that is universal in the limit, i.e.,*

$$\mathcal{F}_{\theta^\star}(x) = y(x), \forall x \in \mathcal{X},$$

*and makes random predictions that are correct with probability*

$$P(\mathrm{maj}(\{\mathrm{sgn}(\langle \boldsymbol{a}_{\theta^\star}^{(j)}, x \rangle - \boldsymbol{b}_{\theta^\star}^{(j)})\}_{j=1}^m) = y(x)) > 1 - \exp\{-2\epsilon^2 m^2\},$$

*where $\epsilon$ is the minimum bias of $\mathcal{F}_{\theta^\star}$.*

*Further, if $p^\dagger$ is the number of parameters used by a deterministic neural network with one hidden layer to achieve zero-error in the task, $f_\theta$ has at most*

$$p \leq p^\dagger + \mathcal{O}(1) \text{ parameters.}$$

*Proof of Theorem 1.* Since Assumption 1 holds[3], there exists a single hidden-layer neural network $N$ that, like $s$, achieves zero-error in this task [8]. Further, since sgn is nonpolynomial, we can use it as the non-linearity of this network [21]. Putting it all together, there exists a number of hidden units $M$ and parameters $b_j, o_j \in \mathbb{R}, w_j \in \mathbb{R}^d$ for $j = 1, \ldots, M$ such that

$$N(x) := \sum_{j=1}^M o_j \mathrm{sgn}(\langle w_j, x \rangle - b_j),$$

and

$$N(x) = \mathrm{sgn}(s(x)), \forall x \in \mathcal{X}.$$

Note that this means we can achieve zero-error in classification, $N(x) = y(x), \forall x \in \mathcal{X}$. Having this in mind, we will now show that for any network $N$ constructed as such, we can build a limiting classifier $\mathcal{F}_\theta$ equal to it. Let us first note that we can force positivity in the output weights of $N$ by doing

$$o_j' := o_j \cdot \mathrm{sgn}(o_j)$$
$$w_j' := w_j \cdot \mathrm{sgn}(o_j)$$

without changing the classification

$$\mathrm{sgn}(\sum_{j=1}^M o_j \mathrm{sgn}(\langle w_j, x \rangle - b_j)) = \mathrm{sgn}(\sum_{j=1}^M o_j' \mathrm{sgn}(\langle w_j', x \rangle - b_j)), \forall x \in \mathcal{X}.$$

Further, note that we can also multiply the output weights by the constant $\sum_{j=1}^M o_j'$ without changing the sign and thus the classification. Overall, we can define a network $N^\star$ equivalent to $N$ in classification with

$$N^\star(x) := \sum_{j=1}^M \frac{1}{\sum_{j=1}^M o_j'} o_j' \mathrm{sgn}(\langle w_j', x \rangle - b_j).$$

Now, note that the limiting classifier can be written as $\mathcal{F}_\theta(x) = \mathrm{sgn}(\mathbb{E}_{\boldsymbol{a}_\theta, \boldsymbol{b}_\theta} \mathrm{sgn}(\langle \boldsymbol{a}_\theta, x \rangle = \boldsymbol{b}_\theta))$. We can then define its distribution as one that samples from the weights and bias $w_j', b_j$ with probabilities according to $\frac{1}{\sum_{j=1}^M o_j'} o_j'$. This implies that $\mathcal{F}_\theta(x) = N^\star(x) = N(x)$, making our $\mathcal{F}_\theta$ achieve zero-error in the task. Note that if $p^\dagger := M$ is the number of parameters in $N$, we can generate the linear coefficients in $\mathcal{F}_\theta(x)$ with at most $p^\dagger + O(1)$ —simply take the external noise from a uniform distribution and map it to the probabilities in the output layer to pick the coefficients.

Finally, note that we have to also characterize the probability that $\mathbf{y}_\theta^{(m)}$ outputs a different answer from $\mathcal{F}_\theta$. The common tool for this is Hoeffding's inequality, which lets us upper-bound it for some $x \in \mathcal{X}$ with

$$P(\mathbf{y}_\theta^{(m)} \neq \mathcal{F}_\theta(x)) \leq \exp\{-2\varepsilon_x^2 m^2\},$$

where $\varepsilon_x = P(\mathbf{y}_\theta^{(m)} = \mathcal{F}_\theta(x)) - 0.5$. Note that $\varepsilon_x$ depends on $x$ and our definition of the minimum bias $\epsilon$ in the main text is precisely the lowest of all such $\varepsilon_x$. Thus, as stated we finally have that for all $x \in \mathcal{X}$

$$P(\mathbf{y}_\theta^{(m)} = \mathcal{F}_\theta(x)) > 1 - \exp\{-2\varepsilon^2 m^2\}.$$

Note that the above is true for any $\mathcal{F}_\theta$, including the universal and invariant ones later discussed in the main text.

$\square$

---

[3]Further taking the usual assumption that $\mathcal{X}$ is compact.

**Theorem 2** (*$G$-invariant RLCs*). *Let $(\boldsymbol{x}, y)$ be a $G$-invariant task with a smooth separator as in Assumption 1. Then, the set of RLCs with a $G$-invariant distribution in the classifier weights, i.e.,*

$$\boldsymbol{a}_\theta \stackrel{d}{=} g \cdot \boldsymbol{a}_\theta, \forall g \in G,$$

*and absolutely continuous randomness source (cf. Assumption 2) is both probabilistic $G$-invariant and universal in $(\boldsymbol{x}, y)$. That is,*

$$\mathcal{F}_\theta(x) = \mathcal{F}_\theta(g \cdot x), \forall x \in \mathcal{X}, \forall g \in G, \forall \theta \in \mathbb{R}^p,$$

*and*

$$\exists \theta^\star \in \mathbb{R}^p : \mathcal{F}_{\theta^\star}(x) = y(x), \forall x \in \mathcal{X}, \forall g \in G.$$

*Proof of Theorem 2.* Let us start with Proposition 3, a central observation needed in Theorem 2. Put into words Proposition 3 says that we can transfer the action of $G$ to the weights of the linear classifier. For the reader more familiar with group representation theory, the result follows immediately from noting that compact groups admit unitary representations, see [28] for a good resource on the matter.

**Proposition 3.** *If $G$ is a compact group and $g \cdot x$ is an action of $g \in G$ on $x \in \mathbb{R}^d$, there exists a bijective mapping $\alpha \colon G \to G$ such that*

$$\langle g \cdot x, w \rangle = \langle x, \alpha(g) \cdot w \rangle, \forall x, w \in \mathbb{R}^d, \forall g \in G.$$

*Proof.* Since we are dealing with actions on a finite-dimensional vector space, the action $\cdot$ of $g$ can be associated with a linear representation $\rho$ of $G$, *i.e.*, $\rho(g) \in \mathbb{R}^{d \times d}$. That is, if $[x]$ is the column vector of $x$, we have that $\rho(g)[x] = g \cdot x$. Now, since $G$ is compact and $\mathbb{R}^d$ is finite-dimensional, $G$ also admits a unitary linear representation $\rho_u$ with associated action $\cdot_u$. Then, it is easy to see that

$$\langle g \cdot_u x, w \rangle = \langle \rho_u(g)[x], w \rangle = \langle x, \rho_u(g)^{-1}[w] \rangle = \langle x, g^{-1} \cdot_u [w] \rangle, \forall w \in \mathbb{R}^d.$$

Thus, since inversion defines a bijection on $G$, Proposition 3 holds for the unitary representation action. Now, note that —as any other group action— $\cdot_u$ defines a homomorphism between $G$ and the symmetric group of $\mathbb{R}^d$. Then, for any other action $\cdot$ there exists a bijective function $\beta \colon G \to G$ such that $g \cdot x = \beta(g) \cdot_u x, \forall x \in \mathbb{R}^d$. Hence, we can write

$$\langle g \cdot x, w \rangle = \langle \beta(g) \cdot_u x, w \rangle = \langle \beta(g)^{-1} \cdot_u x, w \rangle = \langle \beta^{-1}(\beta(g)^{-1}) \cdot x, w \rangle, \forall x, w \in \mathbb{R}^d, \forall g \in G.$$

Finally, since $\beta$ and group inversion are bijective, we finish the proof by defining the bijective mapping $\alpha(g) := \beta^{-1}(\beta(g)^{-1})$ for every $g \in G$. $\qquad \square$

Now, we can proceed to prove the universality part of Theorem 2. Since the task admits a smooth separator, there exists a universal limiting classifier $\mathcal{F}_\theta$ for it. That is, for every $x \in \mathcal{X}$

$$\mathcal{F}_\theta(x) = y(x).$$

Since the task is $G$-invariant, we know that for every $g \in G$

$$\mathcal{F}_\theta(x) = y(g \cdot x).$$

Note that since $G$ is compact, it admits a unique normalized Haar measure $\lambda$[4]. Then, the insight comes from considering a random group action $\mathbf{g} \sim \lambda$. By Fubini's theorem and Proposition 3, we have

$$\mathcal{F}_\theta(x) = \mathbb{E}_\mathbf{g}\big[\mathcal{F}_\theta(\mathbf{g} \cdot x)\big] = \mathrm{sgn}\big(\mathbb{E}_{\mathbf{g}, \mathbf{a}_\theta, \mathbf{b}_\theta}\big[\mathrm{sgn}(\langle x, \alpha(\mathbf{g}) \cdot \mathbf{a}_\theta \rangle - \mathbf{b}_\theta)\big]\big).$$

Now, we can define a limiting classifier $\mathcal{F}_{\theta^\star}$ that samples coefficients according to $\mathbf{b}_{\theta^\star} \stackrel{d}{=} \mathbf{b}_\theta$ and $\mathbf{a}_{\theta^\star} \stackrel{d}{=} \mathbf{g} \cdot \mathbf{a}_\theta$. From above, we know that $\mathcal{F}_{\theta^\star}(x) = \mathcal{F}_\theta(x), \forall x \in \mathcal{X}$. Since $\mathcal{F}_\theta$ is universal, $\mathcal{F}_{\theta^\star}$ is also universal. Finally, since $\mathbf{g}$ is sampled according to the Haar measure, we have that

$$\mathbf{a}_{\theta^\star} \stackrel{d}{=} g \cdot \mathbf{a}_{\theta^\star}, \forall g \in G.$$

At last, it is easy to see from Proposition 3 that the action on the weights can be transferred to the input as well, meaning that every model is probabilistic invariant, *i.e.*, $\mathcal{F}_\theta(x) = \mathcal{F}_\theta(g \cdot x), \forall x \in \mathcal{X}, \forall g \in G, \forall \theta \in \mathbb{R}^p$, which finalizes the proof. $\qquad \square$

**Proposition 1** (*Infinitely $G$-invariant RLCs*). *Let $(\boldsymbol{x}, y)$ be a $G$-invariant task with infinitely $G$-invariant data (Assumption 3) with a smooth separator as in Assumption 1. Then, the set of RLCs with an infinitely $G$-invariant distribution in the linear classifier weights, i.e., as in Assumption 3 $(\boldsymbol{a}_{\theta i})_{i=1}^\infty \stackrel{d}{=} g_\infty \cdot (\boldsymbol{a}_{\theta i})_{i=1}^\infty, \forall g_\infty \in G_\infty$, where $\boldsymbol{a}_\theta \stackrel{d}{=} (\boldsymbol{a}_{\theta i})_{i \in S}$, for $S \in \binom{\mathbb{N}}{d}$, and absolutely continuous randomness source (cf. Assumption 2) is probabilistic $G$-invariant and universal for $(\boldsymbol{x}, y)$ as in Theorem 2.*

---

[4]The reader can think of $\lambda$ as a uniform distribution over $G$.

*Proof of Proposition 1.* From Proposition 3 it also follows that the model is $G$-invariant. Let us proceed to prove it is universal. Since the task admits a smooth separator, from Theorem 1 there exists an RLC $\mathcal{F}_\theta$ such that

$$\mathbb{E}_{\mathbf{x}}[\mathcal{F}_\theta(\mathbf{x}) - y(\mathbf{x})] = 0.$$

Since the data is infinitely $G$-invariant, we have that for the infinite $G$-invariant sequence $(\mathbf{x}_i)_{i=1}^\infty$

$$\mathbb{E}_{\mathbf{x}}[\mathcal{F}_\theta(\mathbf{x}) - y(\mathbf{x})] = \mathbb{E}_{(\mathbf{x}_i)_{i\in S}}[\mathcal{F}_\theta((\mathbf{x}_i)_{i\in S}) - y((\mathbf{x}_i)_{i\in S})] = 0, \text{ for } S \in \binom{\mathbb{N}}{d},$$

with $\binom{\mathbb{N}}{d}$ being the set of all $d$-size subsets of $\mathbb{N}$. Let $\lambda$ be the unique normalized Haar measure of $G$. As in the proof of Theorem 2, we can take a random group action $\mathbf{g} \sim \lambda$ and have from Proposition 3 and Fubini's theorem that

$$\mathbb{E}_{(\mathbf{x}_i)_{i\in S}}[\mathcal{F}_\theta((\mathbf{x}_i)_{i\in S}) - y((\mathbf{x}_i)_{i\in S})] = \mathbb{E}_{((\mathbf{x}_i)_{i\in S})_{i\in S},\mathbf{g}}[\mathcal{F}_\theta(\mathbf{g} \cdot ((\mathbf{x}_i)_{i\in S}))]$$
$$= 0, \text{ for } S \in \binom{\mathbb{N}}{d}.$$

Now, let us define a task $(\mathbf{x}', y)$ on a higher dimension $d' \geq d$, $\mathrm{supp}(\mathbf{x}) \subset \mathrm{supp}(\mathbf{x}') \subseteq \mathbb{R}^{d'}$ where $y(\mathbf{x}') = y(\mathbf{x}'_{[1:d]})$. Further, $\mathbf{g}'$ is a random group action (sampled from the Haar measure) from $G'$, the homomorphism of $G$ into the dimension $d'$. From above, together with Proposition 3 we have that

$$\mathbb{E}_{(\mathbf{x}_i)_{i\in S'}}[\mathcal{F}_\theta((\mathbf{x}_i)_{i\in S'}) - y((\mathbf{x}_i)_{i\in S'})] = \mathbb{E}_{((\mathbf{x}_i)_{i\in S'})_{i\in S'},\mathbf{g}'}[\mathcal{F}_\theta(\mathbf{g}' \cdot ((\mathbf{x}_i)_{i\in S'}))]$$
$$= \mathbb{E}_{((\mathbf{x}_i)_{i\in S'})_{i\in S'}}[\mathrm{sgn}(\mathbb{E}_{\mathbf{a}_\theta,\mathbf{b}_\theta,\mathbf{g}'}[\mathrm{sgn}(\langle x, \alpha(\mathbf{g}') \cdot \mathbf{a}'_\theta\rangle - \mathbf{b}'_\theta)])]$$
$$= 0, \text{ for } S' \in \binom{\mathbb{N}\backslash\{1,\ldots,d\}}{d'}.$$

Now, from above, since the lower dimensions $\{1,\ldots,n\}$ are $G$-invariant if all higher dimensions are $G$-invariant, it is easy to see that there is an infinitely $G$-invariant sequence of weights $(\mathbf{a}_{\theta i})_{i=1}^\infty$ that achieves zero error. $\square$

**Proposition 2** (Universality and resource consumption of RSetCs). *Let $(\mathbf{x}, y)$ be a permutation-invariant task with a smooth separator as in Assumption 1 and infinitely permutation-invariant data as in Assumption 3. Then, RSetCs as in Definition 1 with absolutely continuous randomness source (cf. Assumption 2) are probabilistic $G$-invariant and universal for $(\mathbf{x}, y)$ (as in Theorem 2). Further, the number of parameters needed by RSetCs in this task will depend only on the smallest finite absolute moments of the weight and bias distributions in the zero-error solution, i.e., the ones given by $f_{\theta_f^\star}(\mathbf{u}, \mathbf{u}_i)$ and $g_{\theta_g^\star}(\mathbf{u}, \mathbf{u}_b)$.*

*Proof of Proposition 2.* The result follows directly from the combination of de Finetti's theorem [9], Kallenberg's noise transfer theorem [16] and the recent results on the capacity of neural networks to generate distributions from noise [34]. Let us outline it in more detail. From de Finetti's theorem [9] we know that any infinite sequence of exchangeable random variables can be expressed by i.i.d. random variables conditioned on a common latent measure. Combining this with Kallenberg's noise transfer theorem we have that the weights and biases can be written as $f(\mathbf{u}, \mathbf{u}_i)$ and $g(\mathbf{u}, \mathbf{u}_b)$ where $f, g$ are measureable maps and the noises are sampled from a uniform distribution. Finally, the work of [34] showed that we can replace the uniform noise with absolutely continuous noise and $f, g$ with a sufficiently expressive ReLU multi-layer perceptron. Moreover, Theorem 2.1 in [34] says that the number of parameters depends only on the output dimension and the smallest finite absolute moments of the distributions. Since the output dimension $k$ is constant for us, it depends solely on the latter and we finalize the proof. $\square$

**Theorem 3.** *Let $(\mathbf{x}, y)$ be a graph isomorphism-invariant task that either i) has a smooth separator as in Assumption 1 or ii) is an inner-product decision graph problem as in Definition 3. Further, the task has infinitely graph isomorphism-invariant data as in Assumption 3. Then, there exists an RGraphC as in Definition 2 with absolutely continuous randomness source (cf. Assumption 2) that is probabilistic $G$-invariant and universal for $(\mathbf{x}, y)$ (as in Theorem 2). Further, the number of parameters of this RGraphC will depend only on the smallest finite absolute moments of its weight and bias distributions, i.e., the ones given by $f_{\theta_f}(\mathbf{u}, \mathbf{u}_i)$ and $g_{\theta_g}(\mathbf{u}, \mathbf{u}_b)$.*

*Proof of Theorem 3.* Note that it suffices to show that there exists an RLC that takes as input graphs of size $d$ and decides correctly tasks in Definition 3 with probability $> 0.5$ —this implies a universal limiting classifier

$\mathcal{F}_\theta$. Then, we can replace the smoothness assumption in Theorem 2 and Proposition 1 with Definition 3 and we will have that there exists an infinitely jointly exchangeable sequence that is universal for tasks as in Definition 3. Finally, we follow Proposition 2's proof by simply replacing de Finetti's with Aldous-Hoover's theorem. Thus, let us now proceed to prove the main part of this theorem, *i.e.*, there exists an RLC that takes as input graphs of size $d$ and decides correctly tasks in Definition 3 with probability $> 0.5$.

Define an RLC that samples the linear coefficients as follows.

- Let $\mathbf{a}_\theta'$ be a random variable such that $\mathrm{supp}(\mathbf{a}_\theta') = S$. It is important to note that we do not require that this random vector to have the same distribution as the input, or any other distribution on $S$. We simply need it to be supported on $S$.

- Now, let $t \sim \mathrm{Bernoulli}(0.5 + \gamma)$ for some $\gamma > 0$. We define $\mathbf{a}_\theta := t \cdot \mathbf{a}_\theta'$ and and $\mathbf{b}_\theta := t \cdot b$. Our random prediction can then be rewritten as $\mathrm{sgn}(t \cdot (\langle \mathbf{a}_\theta', x \rangle - b))$.

Let us now calculate our probability of success. For convenience, let us define $\mathrm{sgn}(0) = 1$[5]. Then, the probability of a positive graph $x^+$, *i.e.*, $y(x^+) = 1$, being classified correctly is

$$P(\mathrm{sgn}(\langle \mathbf{a}_\theta, x^+ \rangle - \mathbf{b}_\theta) = 1) = P(\mathrm{sgn}(t \cdot (\langle \mathbf{a}_\theta', x^+ \rangle - b)) = 1)$$
$$= 0.5 - \gamma + P(\langle \mathbf{a}_\theta', x^+ \rangle \geq b) \cdot (0.5 + \gamma),$$

while the probability of a negative graph $x^-$, *i.e.*, $y(x^-) = -1$, being classified correctly is

$$P(\mathrm{sgn}(\langle \mathbf{a}_\theta, x^- \rangle - \mathbf{b}_\theta) = 1) = P(\mathrm{sgn}(t \cdot (\langle \mathbf{a}_\theta', x^- \rangle - b)) = 1) = 0.5 + \gamma.$$

Then, since $\mathcal{X}$ is a finite set for any input distribution $\mathbf{x}$ there exists a constant $\gamma$ where $0 < \gamma/(0.5 + \gamma) < P(\langle \mathbf{a}_\theta', x \rangle \geq b)$ for all $x \in \mathcal{X}$. Given such gamma, every input has a probability of success greater than $0.5$, which implies that $\mathcal{F}_\theta(x) = y(x), \forall x \in \mathcal{X}$ as we wanted to show. $\qquad \square$

## B    RLCs for spherical data

Here we want to show how the same ideas presented in Sections 3.1 and 3.2 can be applied to spherical data using Freedman's theorem [13]. More specifically, we let our input be $d$-dimensional vectors, *i.e.*, $\mathrm{supp}(\mathbf{x}) \subseteq \mathbb{R}^d$ and our task invariant to the orthogonal group, *i.e.*, $y(x) = y(g \cdot x), \forall x \in \mathrm{supp}(\mathbf{x}), \forall g \in G$, where $G := O(d)$. We refer to tasks like this as tasks over spherical data since it is invariant to rotations and reflections.

Just as de Finetti, Aldous and Hoover informed us about sets and graphs, Freedman did it for spherical data [13]. It follows from his work that an infinite sequence invariant to the action of the orthogonal group can be represented as a random scalar multiplying a sequence of i.i.d. standard Gaussian distributions. Leveraging this, we can define the following very simple invariant model for spherical data (assuming Assumption 3).

**Definition 4** (Randomized Sphere Classifiers). *A Randomized Sphere Classifier (RSphereC) uses* 2 *neural networks* $f_{\theta_f} : \mathbb{R}^2 \to \mathbb{R}$ *and* $g_{\theta_g} : \mathbb{R}^2 \to \mathbb{R}$ *together with* 3 *sources of randomness:* $\mathbf{u}, \mathbf{u}_a$ *and* $\mathbf{u}_b$. *The random linear classifier coefficients are generated with*

$$\mathbf{a}_\theta \overset{d}{=} f_{\theta_f}(\mathbf{u}, \mathbf{u}_a) \cdot [\mathcal{N}(0,1), \dots \mathcal{N}(0,1)], \text{ and } \mathbf{b}_\theta \overset{d}{=} g_{\theta_g}(\mathbf{u}, \mathbf{u}_b),$$

*where* $[\mathcal{N}(0,1), \dots \mathcal{N}(0,1)]$ *is a vector of d i.i.d. standard Gaussians.*

**Proposition 4** (Universality and resource consumption of RSphereCs). *Let* $(\mathbf{x}, y)$ *be a* $O(d)$-*invariant task with a smooth separator as in Assumption 1 and infinitely spherical data as in Assumption 3. Then, RSphereCs as in Definition 4 with absolutely continuous randomness source (cf. Assumption 2) are probabilistic $G$-invariant and universal for* $(\mathbf{x}, y)$ *(as in Theorem 2). Further, the number of parameters needed by RSphereCs in this task will depend only on the smallest finite absolute moments of the weight and bias distributions in the zero-error solution, i.e., the ones given by* $f_{\theta_f^\star}(\mathbf{u}, \mathbf{u}_i)$ *and* $g_{\theta_g^\star}(\mathbf{u}, \mathbf{u}_b)$.

*Proof.* The proof is exactly as the one from Proposition 2 but replacing de Finetti's theorem with Freedman's [13]. For a clean and more accessible material on Freedman's theorem, we refer the reader to the lectures by Kallenberg in `https://mysite.science.uottawa.ca/givanoff/wskallenberg.pdf`. $\qquad \square$

Finally, we note that tasks invariant to $O(n)$ are not that common. However, tasks invariant to $SO(n)$ are extremely relevant in computer vision, see [6]. Therefore, we think this model is a first step towards probabilistic invariance for $SO(n)$[6].

---

[5]This can be done by always adding a sufficiently small constant to the input.

[6]Note that $SO(n)$ is the connected component of $O(n)$.

## C    Implementation Details

As mentioned in the main text, all models were trained for a maximum of 1000 epochs using a cosine annealing scheduler for the learning. We tuned all the hyperparameters on the validation set using a patience of 30. The Deep Sets models found a better learning rate of 0.001 and batch size of 250. The GNN model found a better learning rate of 0.01 and batch size 100. The RSetC model used a batch size of 250 and learning rate 0.5. The RGraphC model used a batch size of 100 and a learning rate of 0.5. Finally, we also note that RLCs can out put different answers given the same weights (due to the random input). Thus, we amplify each mini-batch with 1000 samples for each model —we noted that this reduces the variance and helps convergence. The amplification size used in test was $m = 10$. Finally, for all RLCs we used a single dimensional standard normal distribution in each of their noise sources.

Source code is available at: `https://github.com/cottascience/invariant-rlcs`

