# OpenReview forum: "Probabilistic Invariant Learning with Randomized Linear Classifiers"
_NeurIPS.cc/2023/Conference — NeurIPS 2023 poster_

### Official Review · Reviewer_iRPF · 2023-07-05

**Soundness:** 3 good
**Presentation:** 4 excellent
**Contribution:** 3 good
**Rating:** 7
**Confidence:** 3

**Summary:**

The authors propose probabilistic linear classifiers (RLCs) that are able to solve relatively general binary classification problems. They derive a trade-off between the probability that sampling of RLCs leads to an accurate prediction and the number of samples to obtain the majority vote of an RLC.
Furthermore, there discuss more detailed results for three invariant learning problems related to inputs in form of sets, graphs, or spherical data.


**Strengths:**

- **Clarity:** The paper is well written and well understandable despite a strong theoretical component.

- **Originality:** The authors propose to use probabilistic binary classifiers instead of deterministic ones to trade-off the number of required trainable parameters with prediction accuracy. A similar idea has been put forth by Sieradzki et al. (ICML22) for a number of interesting examples. This work seems to be the first to use randomized models for invariant learning (and relatively general binary classification problems).

- **Significance:** The idea is significant because it discusses a novel problem solution perspective to overcome some computational barriers of the deterministic setting.

- **Quality:** Mathematically rigorous analysis and exposition of the main idea. As a highlight, Theorem 1 gives a practically relevant estimate of the trade-off between required samples and the probability of accurate predictions.
The experiments could be designed in better support of the theory but still point out some potential benefits of RLCs for binary classification problems on sets and graphs.

**Detailed points of strengths:**

- The number of parameters used by RSetC does not depend on the set size, in contrast to DeepSets. Yet, the resource consumption depends on the smoothness of the distribution of linear classifier weights and bias. (Note that this might also imply disadvantages over DeepSets in some settings.)

- RGraphCs can approximate any problem that either has a smooth boundary or can be tested with an inner product as in Definition 3. This includes the classification of graph connectivity (which cannot be solved by simple GNN architectures).



**Weaknesses:**

- The analysis is limited to binary classification problems.

- The claimed computational resource savings are not obvious from the exposition and not directly covered by the theorems. According to the theory, the number of required parameters of RLCs could be higher than the ones of a deterministic solution (by a constant that could be very large). The theory does not establish that a smaller number of samples m (leading to potentially lower accuracy) would work in combination with fewer neural network parameters.
(Some potential benefits for invariant learning are established based on reduced input dependence. But the changed criteria related to the smoothness of decision boundaries could also lead to worse results for RLCs in some situations.)

- The experiments were not designed clearly to highlight computational advantages but to show accuracy improvements on small scale tasks, in which the baselines cannot perform reasonably well because of an unrealistic computational resource constraint.
It is likely that the baselines would actually achieve much higher accuracy with sufficient model capacity. Could RLCs actually perform at least on par if equipped with the same capacity that allow the baselines to work well?

- The statement in Line 338: "This comes from the fact that GNNs cannot distinguish simple non-isomorphic graphs such as any pair of d-regular graphs. As such, GNNs cannot universally approximate tasks that assign different labels to any of such pairs. In contrast, RGraphCs can approximate any problem that either has a smooth boundary or can be tested with an inner product..."
is not entirely correct. This limitation only applies to simple GNN architectures. Modern architectures that use attention and suitable node labels do not suffer from this problem. Experiments that compare with such stronger baselines would be more informative regarding the practical implications of the proposed work.

- Conceptually, RLCs might require fewer trainable parameters in some cases, which could lead to advantageous memory requirements. Yet, they are not necessarily computationally more efficient at inference, which is not discussed at all in the paper. The fact that each inference needs to evaluate the trained neural network m times could increase the associated FLOPS significantly.

- For the same reason, the experiments might not be fair, as the baselines are allowed fewer computational resources at inference than the RLCs.

- **Reproducibility:** No code has been shared.


**Points of minor critique:**

- Line 297: $u_b$ is listed twice instead of once.

- Figures: It would help to color also confidence intervals because it currently is impossible to distinguish them when they are overlapping.

- The size of the validation set is not mentioned (neither in the main paper nor the Appendix). Only the size of the training and test set is given.


**Questions:**

- How many samples m did RsetC and RgraphC use in the experiments? This seems to be quite an important parameter to determine the maximum achievable accuracy and the required computational resources for the evaluation of RsetC and RgraphC.

- Experimental set-up in support of story:
I would like to understand the required computational resources in an experimental context that directly supports the theoretical claims. 1) Start from good baselines that have enough neural network capacity to actually solve the discussed problems. 2) Then design RLCs that achieve a similar accuracy and compare the used computational resources. In this context: How many FLOPS does the evaluation of RLCs cost? How many trainable parameters do they use?


**Limitations:**

The limitations have been pointed out under weaknesses and were partially discussed by the authors. I do not foresee a major or immediate negative societal impact of this work.

---

> ### Author Rebuttal · Authors · 2023-08-08
>
> Thank you for the valuable comments. We are glad the reviewer appreciated our contributions. In the feedback summary we address your questions about i) settings where the deterministic models succeed and how it can be compared to RLCs in terms of resources and ii) how the number of samples $m$ can impact our resources. We will also take in your suggestion about color coding in the plots for the final version.
>
> >  According to the theory, the number of required parameters of RLCs could be higher than the ones of a deterministic solution (by a constant that could be very large).
>
> In the proof of Theorem 1 we can see that the constant cannot be arbitrarily large. If $p^\dagger$ is the number of parameters that an MLP with 1 hidden layer needs to solve the task, the constant will be at most 2. We thank the reviewer for pointing out this is not clear in the main text and we will make it explicit in the final version.
>
> > The statement in Line 338: [...] is not entirely correct.
>
> Thanks, we will clarify that we refer to GNNs with expressiveness bounded by the Weisfeiler Leman test. Note, however, that the mentioned node labeling (e.g., random features) methods do not have the invariance property, the main theme of our work.
>
> > "How many samples m did RsetC and RgraphC use in the experiments?
>
> 10, we will add to the main paper in the final version.

---

> > ### Comment · Reviewer_iRPF · 2023-08-14
> > **Estimate of FLOPs**
> >
> > I thank the authors for the clarifications.
> >
> > Could they also report the FLOPs associated with their methods in comparison with the baseline? The information that is currently provided suggests that baseline models with sufficient capacity can achieve a higher accuracy potentially. However, how big are the achieved computational savings exactly?
> > Or could the accuracy of the proposed methods be boosted by increasing the number of samples?

---

> > > ### Author Response · Authors · 2023-08-15
> > > **FLOPs**
> > >
> > > We thank the reviewer for the interest in our work.
> > >
> > > > Comparison of FLOPs.
> > >
> > > This is a great question. Our theory shows that our models can achieve universality with a number of parameters independent of the maximum size of the input sets and graphs---an important distinction between RLCs and deterministic classifiers, because these maximum sizes are not typically known in practice. But, as you suggest, we cannot provide a guarantee on the time complexity (or number of FLOPs), and certainly one could design settings in which RLCs need many samples to provide accurate predictions with high probability.
> > >
> > > To address your question empirically, we've re-run our in-distribution experiments using a DeepSets model that uses at least as many FLOPs as the RLCs. Since we cannot upload pdfs in the comments, we write in a separate comment a table with results. As you can see the results are similar to the results in the paper (with RLCs dominating at large set sizes). Thanks for asking this question; we will include this result in the final draft.
> > >
> > > > The information that is currently provided suggests that baseline models with sufficient capacity can achieve a higher accuracy potentially.
> > >
> > > This is certainly true for Deep Sets when the hidden layer size matches the maximum input set size, but it's never true for GNNs (see line 338). Our benefits in set tasks are in the regime of a constant number parameters.
> > >
> > > > Or could the accuracy of the proposed methods be boosted by increasing the number of samples?
> > >
> > > Certainly it can increase, but as we noted in Theorem 1 and in the feedback summary, the predictions converge exponentially with the number of samples, thus this benefit tends to be limited by a few samples ---due to our sample efficiency.
> > >
> > > We thank the reviewer once again for their valuable feedback, we will add this discussion to the final version.

---

> > > > ### Author Response · Authors · 2023-08-15
> > > > **mentioned results**
> > > >
> > > > |   Set size |   Deep Sets (hidden size = 50) |   RSetCs (hidden size = 5) |
> > > > |-----------:|-------------------------------:|---------------------------:|
> > > > |          5 |                          0.6   |                      0.618 |
> > > > |          9 |                          0.59  |                      0.614 |
> > > > |         13 |                          0.64  |                      0.658 |
> > > > |         17 |                          0.588 |                      0.634 |
> > > > |         21 |                          0.612 |                      0.682 |
> > > > |         25 |                          0.53  |                      0.528 |
> > > > |         29 |                          0.562 |                      0.602 |
> > > > |         33 |                          0.528 |                      0.518 |
> > > > |         37 |                          0.542 |                      0.57  |
> > > > |         41 |                          0.6   |                      0.628 |
> > > > |         45 |                          0.562 |                      0.586 |
> > > > |         49 |                          0.502 |                      0.55  |
> > > > |         53 |                          0.498 |                      0.59  |
> > > > |         57 |                          0.49  |                      0.63  |
> > > > |         61 |                          0.478 |                      0.498 |
> > > > |         65 |                          0.544 |                      0.596 |
> > > > |         69 |                          0.556 |                      0.588 |
> > > > |         73 |                          0.526 |                      0.562 |
> > > > |         77 |                          0.544 |                      0.562 |
> > > > |         81 |                          0.506 |                      0.508 |
> > > > |         85 |                          0.496 |                      0.492 |
> > > > |         89 |                          0.5   |                      0.554 |
> > > > |         93 |                          0.498 |                      0.536 |
> > > > |         97 |                          0.514 |                      0.554 |
> > > > |        101 |                          0.498 |                      0.614 |

---

### Official Review · Reviewer_4yWn · 2023-07-06

**Soundness:** 3 good
**Presentation:** 3 good
**Contribution:** 3 good
**Rating:** 6
**Confidence:** 3

**Summary:**

The paper introduces a novel approach for achieving universality and invariance in binary classification tasks, while minimizing computational requirements. Instead of relying on deterministic neural networks such as DeepSet, which have parameterization complexity proportional to the set size, the paper proposes using randomised linear classifiers (RLCs). These RLCs can maintain invariance to compact group transformations and provide a universal approximation. Importantly, the parameterization complexity of RLCs remains independent of the set size. Building upon this finding, the paper extends the design of RLCs to ensure invariance in classification tasks involving sets, graphs, and spherical data. Experimental results demonstrate that the proposed RLCs outperform DeepSet and GNN approaches in certain invariant tasks.

**Strengths:**

•	The paper makes a notable theoretical contribution by presenting a novel approach that utilizes randomness to achieve universality and invariance in binary classification tasks. As far as my knowledge extends, this is the first approach of its kind to utilize randomness in this context.

•	The incorporation of de Finetti's, Aldous-Hoover's, and Freedman's theorems is a significant contribution of the paper. These theorems play a crucial role in making RLCs applicable and practical for dealing with set, graph, and spherical data. By leveraging these theoretical foundations, the paper provides valuable insights into achieving invariance and universality across various data domains. This expands the scope of invariance and opens up new possibilities for addressing similar challenges in different contexts.


**Weaknesses:**

My main concern pertains to the scalability of the proposed approach. While deterministic neural networks only require a single feed-forward process to produce an output, RLCs necessitate the sampling of multiple randomnesses to compute the linear weights. This introduces computational complexity during both training and testing phases. Consequently, I am particularly interested in understanding how RLCs perform on real-world tasks compared to deterministic models like DeepSet and GNN. It remains to be seen whether RLCs can deliver competitive results in practical scenarios, considering the additional computational demands imposed by their inherent stochastic nature.

**Questions:**

•	What impact does the hyperparameter $m$ have on the model's performance? It appears that a larger value of $m$ is necessary to achieve highly confident predictions. This suggests that increasing $m$ allows the model to gather more evidence and make more precise decisions. It would be valuable to investigate how different values of $m$ influence the model's accuracy and confidence levels across various tasks and datasets.

•	The impact of random distribution on model performance is a noteworthy aspect to consider. I am curious about whether different randomness would affect the model performance. What kind of randomness do you use in the experiment?


**Limitations:**

The paper presents some notable limitations despite its promising theoretical results:

•	Scalability: The applicability of the proposed RLCs appears to be challenging in large-scale scenarios compared to deterministic methods. Further investigations and enhancements are needed to improve scalability and address potential computational complexities.

•	Unclear impact of $m$ and randomness: The paper would benefit from additional discussions and experiments to explore the effects of the hyperparameter $m$ and the selection of randomness sources. This would provide better insights into the optimal choices and their influence on model performance.

•	Lack of real-world tasks: The evaluation of the proposed method is limited to synthetic tasks, and it would be valuable to extend it to real-world datasets. This would allow for a more comprehensive understanding of the method's capabilities and potential applications.

---

> ### Author Rebuttal · Authors · 2023-08-08
>
> Thank you for the review! In the feedback summary we address your questions about $m$ and the choice of randomness source. Please, let us know if you have any extra input, we'd be interested in discussing.

---

### Official Review · Reviewer_QYFX · 2023-07-06

**Soundness:** 3 good
**Presentation:** 3 good
**Contribution:** 3 good
**Rating:** 7
**Confidence:** 4

**Summary:**

This work presents a very interesting method for training efficient invariant classifiers by leveraging randomness. More precisely, it proposes to train a neural network to sample linear classifier weights, by pushing forward some data-independent distribution, and using the majority vote over sampled classifiers to make predictions. It is rather a theoretical paper which proves universal approximation and G-invariance theorems for this class of classifiers (adapted to the probabilistic setting). The special cases of set and graph invariance are studied (with relaxed assumptions). Spherical data is also addressed in the appendix. A few toy experiments illustrate the theoretical results proved.

**Strengths:**

### Originality

Although the paper borrows some ideas from CFNNs, it is largely original in its restriction to linear classifiers and focus on invariant representation learning.

### Clarity

The paper is clearly written and well-structured.

### Quality

I read two out of three proofs from the supplementary material and they look fine to me. Toy experiments are well-designed.

### Significance

This work addresses the important question of invariant representation learning with a very original method that has some benefits compared to existing methods.

**Weaknesses:**

### Originality

1. The only remark I would have here is that there are a few references missing from the invariant representation learning paragraph of the related work section. Namely, there are methods which are quite different from the proposed one but which also make use of randomness to obtain invariant representations, such as Augerino (https://arxiv.org/abs/2010.11882) and AugNet (https://arxiv.org/abs/2202.02142).

### Clarity

2. The only remark I have is that the error bars on the figures are difficult to associate to each dot as they are not color-coded.

Typos:

- l.126: supp(x)
- l. 297: “$u_b$, and $u_b$” ($u_b$ twice)

### Quality

I only have two remarks/questions regarding the method and experiments:

3. I realize that the paper contribution is rather theoretical, but the experiments are very very toy. I realize the value of the experiments carried out, but I wonder whether it would be possible to showcase the method in a more realistic setting.
4. As far as I understood, the model invariance relies on underlying theorems specific to each setting (de Finetti for sets, Aldous-Hoover for graphs and Freedman for spherical data). Is that right? It would be interesting if you could discuss to which extent your framework can be adapted to arbitrary symmetry groups and whether this is a limitation.




**Questions:**

See questions above.

**Limitations:**

A few limitations are discussed throughout the paper, but not very thoroughly. A major limitation for now is that this is a very preliminary study showcasing the new method only on toy binary classification problems. Although one cannot expect it to address all research questions in a single paper, some directions on how the proposed approach could have practical use would be welcome.

---

> ### Author Rebuttal · Authors · 2023-08-08
>
> Thank you very much for your feedback. We refer the reviewer to our feedback summary for an additional experiment. We will also take in your suggested references in the final version. If you have any other input we'd be happy to discuss.
> > "As far as I understood, the model invariance relies on underlying theorems specific to each setting (de Finetti for sets, Aldous-Hoover for graphs and Freedman for spherical data). Is that right? It would be interesting if you could discuss to which extent your framework can be adapted to arbitrary symmetry groups and whether this is a limitation."
>
> Theorem 2 shows that our framework can be adapted to any compact group transformation. That is, there is an equivalence between designing invariant distributions and designing invariant RLCs. Any result characterizing distributions invariant to a compact group transformation, such as the ones you cited, can be used to design invariant RLCs. We will highlight this in the final version, thank you for bringing this up.

---

> > ### Comment · Reviewer_QYFX · 2023-08-17
> > **Answer to rebuttal**
> >
> > I would like to thank the authors for their thoughtful rebuttal. I have read it, as well as the discussions with authors reviewers. Most of my concerns have been well addressed and I am hence raising my rating by one point.

---

### Official Review · Reviewer_2JHn · 2023-07-10

**Soundness:** 3 good
**Presentation:** 3 good
**Contribution:** 2 fair
**Rating:** 4
**Confidence:** 3

**Summary:**

The authors introduce Randomized Linear Classifiers, which are a way to randomly represent the weights of a linear classifier. Because there is a random reconstruction of the network, a sample majority of which is used to determine the actual inference of the network. The authors show a universal representation theorem for such a classifier, and show that the representation is quite efficient in some cases (for instance, sometimes the number of parameters needed does not scale with a relevant quantity, in which a deterministic network would scale).

**Strengths:**

The notion that the number of parameters needed for $f_\theta$ can be much smaller than the number of parameters needed for a classifier that is a deterministic neural network is nice.

**Weaknesses:**

From what I can tell, the paper only considers the expressivity of RLCs, and the number of parameters needed. Is there a result about learning $f_\theta$?

**Questions:**

Typos in line 219, 287.

As a sanity check that I have understood this, you dont have access to the true gradients for $\theta$ until $m \to \infty$ right?

Is this similar to learning a one layer NN in which during training and inference some random subset of neurons is used each time (for forward and backward passes)?

---

> ### Author Rebuttal · Authors · 2023-08-08
>
> Thank you for reviewing our paper. Next, we address your comments. Please, let us know of any additional feedback you might have. We would be very happy to discuss.
>
> > "From what I can tell, the paper only considers the expressivity of RLCs, and the number of parameters needed. Is there a result about learning $f_{\theta}$?
>
> Our work address the well known computational resources vs. model capacity tradeoff in invariant models. We note that even in deterministic invariant learning the question of generalization/learnability is still to be understood, see for instance https://openreview.net/forum?id=HxeTEZJaxq
> We believe that our probabilistic view on the problem can, in a future work, shed new light into the generalization capabilities of invariant models.
>
> > "As a sanity check that I have understood this, you dont have access to the true gradients for $\theta$ until $m \to \infty$ right?"
>
> Yes, we have gradient samples.
>
> > "Is this similar to learning a one layer NN in which during training and inference some random subset of neurons is used each time (for forward and backward passes)?"
>
> Yes, this is precisely the idea behind the proof of Theorem 1 as you can see in the appendix.

---

### Official Review · Reviewer_EYc5 · 2023-07-26

**Soundness:** 3 good
**Presentation:** 3 good
**Contribution:** 3 good
**Rating:** 5
**Confidence:** 2

**Summary:**

This work proposes a framework called Randomized Linear Classifiers (RLCs) that leverages external randomness to build models that are expressive and can encode invariance in the input space. The authors establish probabilistic versions of universal approximation theorem and invariance for several types of RLCs. The key insight from the theory is that by maintaining probabilistic universal approximation and invariance, RLCs can be more parameter-efficient than their deterministic counterparts. Numerical experiments verify the theoretic properties.

**Strengths:**

- The paper is generally well-written and explains the implications of the assumptions and theorems clearly.
- RLCs extend the idea of Coin-Flipping Neural Network with stronger theoretic backgrounds and more general formulations and invariant extensions.
- The probabilistic notions of universal approximation and invariance potentially open up a new direction.

**Weaknesses:**

- I am not so sure about the benefits of "Online computation" and "Private computation". What is the difference between standard online inference and RLCs? In the inference phase, the client downloads the model and computes the prediction without sending the inputs, so it is already private.
- I am not sure how the external randomness would affect the generalization performance of RLCs. From the theorems, it seems that the only requirement for the randomness source is to be absolutely continuous, thus MLPs can universally approximates? In the experiments, the external randomness was fixed as standard normal. I assume that the distribution of the external randomness would greatly affect sample complexity? For example, RLCs may require more restarts to amplify the probability if a bad external randomness is used? I think this part needs some further discussion.
- In the numerical experiments, the performance of DeepSets looks pretty bad, even worse than using a constant predictor. Is this normal? It would be more convincing if RLCs was examined on tasks that DeepSets performs good.


**Questions:**

It would be interesting to see how the external randomness affects the generalization performance of RLCs.

---

> ### Author Rebuttal · Authors · 2023-08-08
>
> We thank the reviewer for their comments. Please, refer to our feedback summary where we address your questions about the randomness distribution and the deep sets comparison. If you have any extra input, we would be very happy to discuss.
>
> >I am not so sure about the benefits of "Online computation" and "Private computation". [...] What is the difference between standard online inference and RLCs?
>
> The benefits are in terms of resource consumption. In standard online inference, one can ideed send (or store) the model, but this can be arbitrarily large. In our case, we simply need to send (or store) the pre-sampled linear coefficients. As we say in the private computation paragraph, this can be very useful in settings with low-resource computers, e.g., smartwatches.

---

### Author Rebuttal · Authors · 2023-08-08

### **_Feedback summary_**
We thank all the reviewers for the valuable feedback on our manuscript. In general, reviewers found the paper to be well written and appreciated the novel direction given by our theoretical contributions. Here we address three common points raised across reviews. Then, we discuss specific comments separately in each reviewer's thread.
1. **Choice of external randomness:** Reviewers asked about the impact of the choice of randomness source. We highlight that our theory shows the suffiency of absolutely continuous sources. As in models such as GANs or VAEs, due to their capacity, neural networks can transport simple distributions, e.g., Gaussian, to arbitrarily complex ones. Our experiments do confirm that this also seems to be the case in RLCs ---note that RLCs solve supervised learning, a task easier than data generation (tackled by GANs and VAEs). Moreover, if there's a form of prior knowledge about the solution, i.e., the linear coefficients ideal distribution, one should leverage it. In fact, this is precisely what our invariant models are doing (through parameter-sharing), see Theorem 2.
2. **Number of samples ($m$):** Some reviewers asked about the impact of the number of samples $m$ in the computational complexity of the model. We point them to Theorem 1, where we show that we need only a few samples to output the model's true prediction with arbitrarily high probability. For instance, if the model outputs 1 with probability 2/3, with 3 samples we will output 1 with probability $>0.95$. This result is only possible due to the independence between input and randomness source.
4. **Experiments:** Reviewers raised two interesting points regarding our experiments. How do RSetCs compare to tasks when Deep Sets are supposedly good? Can we compare the resource consumption of RSetCs vs. Deep Sets when both are performing well in a task? To address them, we consider the task of deciding whether the majority of a set of random numbers is positive or negative. This task is essentially removing the sorting operation from the task in the paper ---where deep sets struggles. We then consider Deep Sets with one and two hidden layers vs an RSetC with a single hidden layer. The results are in the image attached. We see that the single layer RSetC has a performance more similar to the Deep Sets with two hidden layers, showcasing our parameter efficiency. The experimental setup was the same as the sort task in the main paper. We will add this experiment to the final version, thank you for the input!

---

### Decision · Program_Chairs · 2023-09-21

**Decision:**

Accept (poster)

**Comment:**

An interesting paper, which saw its ratings increase at discussion phase. I am inclined to remove the bottom review of 2JHn, which was short and without any follow-up.

We end up with a paper with globally positive ratings, and that could reasonably be considered for acceptance.